# Gap junctions allow transfer of metabolites between germ cells and somatic cells to promote germ cell growth in the *Drosophila* ovary

**Caroline Vachias, Camille Tourlonias, Louis Grelée, Nathalie Gueguen, Yoan Renaud, Parvathy Venugopal, Graziella Richard, Pierre Pouchin, Emilie Brasset, Vincent Mirouse** [ID]*

Université Clermont Auvergne, Institute of Genetics, Reproduction and Development (iGReD), UMR CNRS 6293—INSERM U1103, Faculté de Médecine, Clermont-Ferrand, France

* vincent.mirouse@uca.fr

## Abstract

Gap junctions allow the exchange of small molecules between cells. How this function could be used to promote cell growth is not yet fully understood. During *Drosophila* ovarian follicle development, germ cells, which are surrounded by epithelial somatic cells, undergo massive growth. We found that this growth depends on gap junctions between these cell populations, with a requirement for Innexin4 and Innexin2, in the germ cells and the somatic cells, respectively. Translatomic analyses revealed that somatic cells express enzymes and transporters involved in amino acid metabolism that are absent in germ cells. Among them, we identified a putative amino acid transporter required for germline growth. Its ectopic expression in the germline can partially compensate for its absence or the one of Innexin2 in somatic cells. Moreover, affecting either gap junctions or the import of some amino acids in somatic cells induces P-bodies in the germ cells, a feature usually associated with an arrest of translation. Finally, in somatic cells, innexin2 expression and gap junction assembly are regulated by the insulin receptor/PI3K kinase pathway, linking the growth of the two tissues. Overall, these results support the view that metabolic transfer through gap junction promotes cell growth and illustrate how such a mechanism can be integrated into a developmental program, coupling growth control by extrinsic systemic signals with the intrinsic coordination between cell populations.

## Introduction

Gap junctions are channels between adjacent cells. Each cells harbor a hemichannel made of six subunits, connexins in vertebrates and innexins in other animal phyla where gap junctions are found. These two protein families are very different in terms of primary sequence, but similar in terms of conformation [1–4]. Hemichannels can be homomeric or heteromeric in composition and channels can be formed from two identical or different hemichannels, leading to a large range of channel combinations with different regulations or solute specificities.

**Data availability statement:** Trap data were deposited on GEO (GSE230452).

**Funding:** This research was financed by the French government IDEX-ISITE initiative 16-IDEX-0001 (CAP 20-25) (to VM). The funder had no role in study design, data collection and analysis, decision to publish, or preparation of the manuscript.

**Competing interests:** The authors have declared that no competing interests exist.

**Abbreviations :** *Cx37*, connexin 37; eIF2α, eukaryotic translation initiation factor 2 subunit alpha; InR/PI3K, Insulin Receptor/Phosphatidylinositol 3 Kinase; Inx4, innexin 4; P-bodies, processing bodies; ROS, reactive oxygen species; S51, serine 51; TOR, Target of Rapamycin; TRAP, translating ribosome affinity purification.

Gap junctions generally assemble in plaques that contain many channels and each channel allows the direct communication, and thus molecular flows, between the cytoplasm of two cells. However, due to the channel size and conformation, only passive transfer of small molecules and ions, up to approximatively 1 kDa, including second messengers (e.g., inositol triphosphate and calcium) is allowed. Gap junction proteins also participate in other cellular mechanisms, such as cell migration [5,6]. Moreover, many molecules that can diffuse through gap junctions are linked to energetic metabolism (e.g., glucose and pyruvate) and to anabolic metabolism (e.g., amino acids). However, despite many studies on these metabolite flows, their functional relevance has been elusive.

Such metabolite exchanges could promote cell growth. Available genetic data suggest that the growth of mammalian oocytes could be an illustration of such a mechanism. Indeed, mutation of connexin 37 (*Cx37*) in germ cells or of *Cx43* in somatic granulosa cells, which surround and are in contact with the oocyte, strongly impairs oocyte and follicle growth [7,8]. Moreover, amino acid import and pyruvate production are more efficient in follicle cells, and granulosa cells are required for an effective uptake of some amino acids, including alanine and proline, by the oocyte [9–12]. These data suggest a potential metabolic flow towards the oocyte via the gap junctions. Nonetheless, it is not known: (i) to which extent this putative transfer of metabolites contributes to explain gap junction impact on oocyte growth; and (ii) whether such mechanism is coupled with follicle cell growth and more generally, is integrated in the genetic programme controlling follicle development.

A major feature of oocytes is their large size throughout animal evolution, despite important variation among species [13,14]. Their size prefigures the early embryo size and their content will determine the early embryo development success rate and quality [15,16]. Hence, oocyte growth is an essential step that requires robust underlying mechanisms [17,18]. Oocyte development and growth usually occur in ovarian follicles where germ cells are surrounded by follicle cells (i.e., somatic epithelial cells). *Drosophila* oogenesis takes place in a structure called ovariole, in which follicles continuously arise and develop from the anterior to the posterior end. Each ovary is subdivided in about 16 ovarioles (Fig 1A). Follicles bud from a structure called the germarium that contains germ cells and somatic stem cells. Follicle development is divided in 14 morphological stages during which the oocyte volume increases 1000 times in about 3 days before being fertilized and laid [19]. At stage 1, a follicle contains a germline cyst of 16 interconnected cells: 15 nurse cells that will grow by endoreplication and 1 oocyte at the posterior end, blocked in early meiosis. Each germline cyst is encapsulated by the follicular epithelium that is initially composed of about 30 cells. These cells proliferate giving 800 cells at stage 6 and then they undergo endoreplication [20]. Both germ cell growth and follicle cell growth are cell autonomously sensitive to the usual growth systemic signals, such as the Insulin Receptor/Phosphatidylinositol 3 Kinase (InR/PI3K) and Target of Rapamycin (TOR) pathways [18,21–23]. These pathways affect also non-cell autonomously the growth of the adjacent tissue in both direction: the soma influences germline growth and vice versa [18,22–25]. The most striking illustration of this reciprocal coordination is observed when *Pten*, a InR/PI3K repressor, is mutated in follicle somatic cells. The faster growth of these somatic cells induces the faster growth of the surrounded wild-type germ cells. Regulation of somatic cell growth by germline growth involves the Hippo pathway and its modulation by tension induced on the epithelium due to the germline cyst volume increase [26]. However, it is not known how the follicle somatic cells control germ cell growth.

Here, we tested the hypothesis of an evolutionary conservation of gap junction function between germ cells and follicle cells to regulate *Drosophila* female germ cell growth and explored the underlying mechanism. Several reports indicate that gap junctions are present during fly oogenesis. In gem cells, innexin 4 (Inx4) forms plaques at the contact with follicle

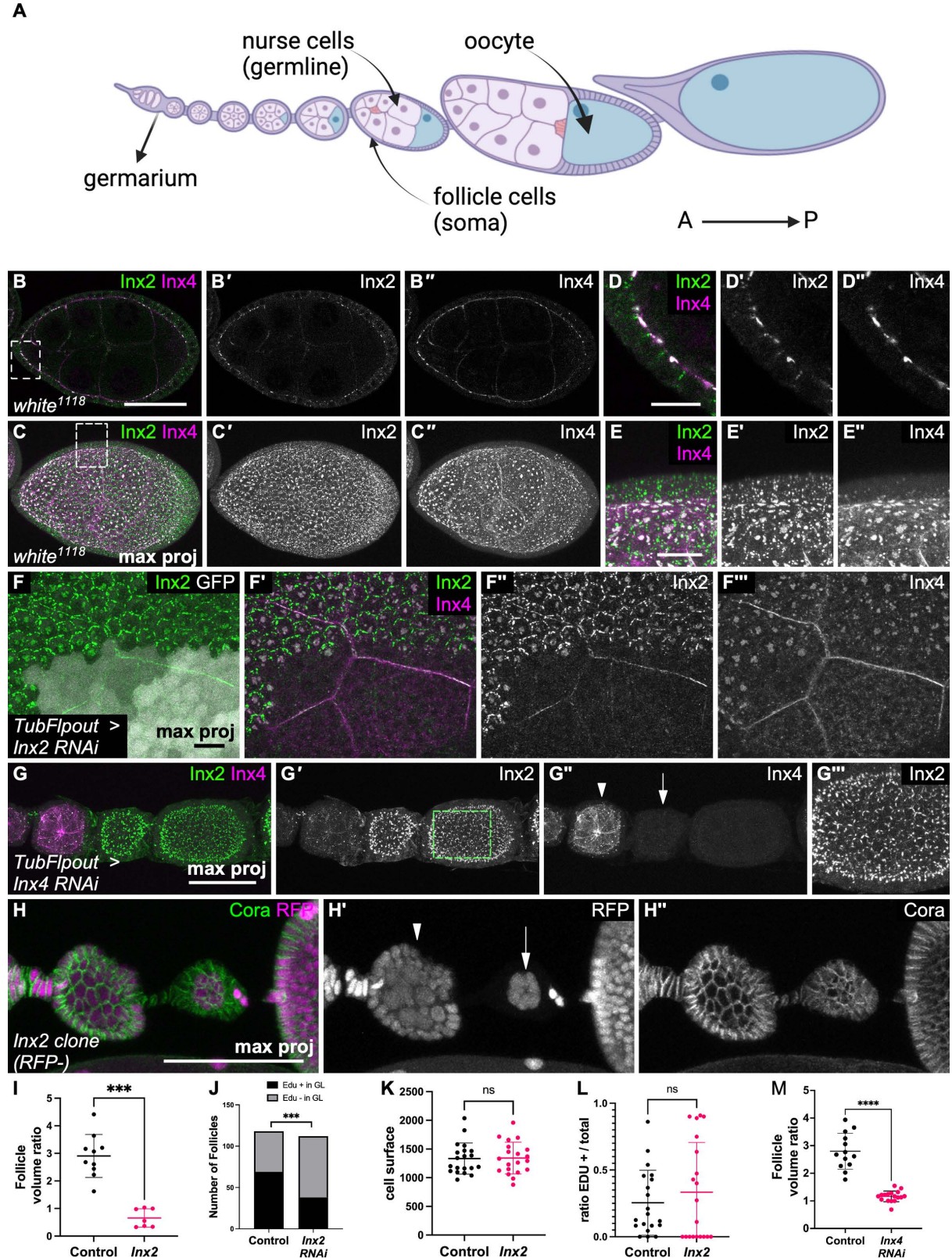

**Fig 1. Soma-germline gap junctions are required for germline growth.** (A) Scheme of an ovariole, oriented from the anterior (A) to the posterior (P) end, as all the subsequent images. The ovariole starts with the germarium from which young follicles bud before undergoing massive growth until the formation of an egg at the posterior end. In each follicle, somatic follicle cells (in purple) surround a germline cyst

with the nurse cells (pink) and the oocyte (in blue). (**B**) Sagittal view and (**C**) maximum intensity projection images of a stage 7 follicle after immunostaining for Inx2 (green in B and C, white in B′ and C′) and Inx4 (magenta in B and C, white in B″ and C″). (**D, E**) Higher magnification of the insets in B and C to illustrate the formation of plaques at the germline-soma interface. (**F**) Immunostaining for Inx2 (green in F and F′, white in F‴) and Inx4 (magenta in F′, white in F″) in a follicle containing a *Inx2* RNAi-expressing clone in follicle cells marked by GFP expression (white in F). (**G**) Maximum intensity projection of immunostaining for Inx2 (green in G, white in G″ and G‴) and Inx4 (magenta in G, white in G′) in follicles containing *Inx4* RNAi-expressing germline clones visualized by Inx4 absence. Note the absence of Inx2 plaques at germline contacts at higher magnification (G‴) and the growth defect of *Inx4* RNAi follicles (arrow) compared with the wild-type younger follicle (arrowhead) (G″). (**H**) Somatic *Inx2*A mutant clones, marked by the absence of RFP expression, induce a germline growth defect when they cover the whole epithelium (arrow) compared with the wild-type younger follicle ($n − 1$) (arrowhead). (**I**) Quantification of the ratio between the volume of the follicle showing a growth defect fully covered by *Inx2* mutant cells, and the volume of the $n − 1$ follicle of the same ovariole ($n = 10$ control and 7 *Inx2* mutant follicles, Mann–Whitney test). (**J**) quantification of stage 1–8 follicles positive or negative for Edu in the germline (GL) ($n = 118$ for control and $n = 112$ for *Inx2* RNAi, Fisher's exact test) (**K, L**) quantification of (K) cell surface and (L) EDU-positive cells in small mutant *Inx2* clones compared with the surrounding wild-type cells ($n = 20$, 20 clones or groups of wild-type cells for K and L, Welch's $t$-test). (**M**) Quantification of the volume ratio between a follicle with *Inx4* RNAi in the germline cyst and the wildtype $n − 1$ follicle of the same ovariole ($n = 13$ for control and $n = 17$ for *Inx4* RNAi, Mann–Whitney test). For all graphs, data are the mean ± SD. ***$p < 0.01$, ****$p < 0.0001$. Scale bars: 50 μm in B, G, H and 10 μm in E, F. The raw data underlying this figure can be found in S1 Data.

cells [27–29]. Several innexins, including Inx2, are expressed in follicle cells and partially colocalize with or are juxtaposed to Inx4 [28,29]. Inx4 is required for germ cell survival before follicle formation, and Inx2 is involved in the germline cyst encapsulation by follicle cells during follicle budding [27,30,31]. Inx2 and Inx4 are also required for morphogenetic events during oogenesis, such as the stretching of anterior follicle cells and the migration of border cells during stage 9 [6,29]. Moreover, Inx4 and Inx2 form gap junctions in the testis where they are required at different steps of sperm development [27,32,33].

We found that Inx4/Inx2 gap junctions are required for female germline growth once follicles are formed. Moreover, follicle cells specifically express genes involved in amino acid biology, and the import of some amino acids in follicle cells is necessary for germline growth. This germline growth control by somatic follicle cells is linked to the formation of processing bodies (P-bodies) and can be partially bypassed by the direct expression of a putative amino acid transporter in the germline. Moreover, gap junction assembly is controlled by the InR/ PI3K pathway, thus connecting growth coordination between these two cell types with systemic growth control.

## Results

### Inx2 and Inx4 form gap junctions at the soma–germline interface that are required for germline growth

As gap junctions are required for oocyte growth in mammals, we asked whether they could play a similar role in *Drosophila* oogenesis [7,8]. Therefore, first, we characterized the gap junction composition and profile in somatic and germ cells, focusing on stages 1–8 when follicle development is globally limited to growth without major morphogenetic changes. As previously described [27–29], we detected Inx2 and Inx4 forming plaques at the interface between germ and somatic cells, these plaques often being larger in follicle anterior part. Inx2 also formed plaques on the lateral domain of follicle cells (Fig 1B–1E). Inx2 expression and plaque size tended to increase as follicles developed (S1A Fig). However, plaques tended to disappear specifically from the oocyte cortex around stage 8 and were completely absent at this position from stage 9 onwards (S1B Fig). The cytoplasmic signal localization suggested that Inx2 and Inx4 were mainly expressed in the soma and germline, respectively. Accordingly, *Inx2* knockdown (*Inx2* RNAi) induction in somatic clones led to the cell-autonomous loss of Inx2

staining, whereas induction of *Inx4* knockdown (*Inx4* RNAi) in the germline led to the loss of Inx4 staining (Fig 1F, 1G). Notably, we did not detect Inx4 at the contact of *Inx2* RNAi follicle cells, confirming published results (Fig 1F‴) [29]. Similarly, when *Inx4* was knocked down in germline cysts, we did not detect Inx2 plaques at the germline contact, although Inx2 protein expression was increased in follicle cells and still localized at the lateral domain of the cells (Fig 1G‴). This last observation could suggest a feedback mechanism between gap junction formation and Inx2 expression. Conversely, the loss of *Inx2* in the germline or *Inx4* in the soma did not induce visible defects (S1C, S1D Fig). Altogether, these data clearly established the presence of gap junction plaques composed of Inx4 in the germline and of Inx2 in follicle cells, and the Inx4-Inx2 interdependence for plaque assembly.

We then tested whether these gap junctions influenced germline growth by generating large *Inx2* mutant clones. In such conditions, we observed follicles that were smaller than the younger one at their anterior, a phenomenon never observed in the wild-type ovarioles. The quantification of the ratio between the volumes of the defective growth follicle and the follicle at its anterior confirmed these observations (Fig 1H, 1I). We observed this phenotype with three different alleles, and only when all (or almost) epithelial cells of a follicle harbored the mutated *Inx2* while the anterior one was wild-type or contained a smaller percentage of mutant cells, whereas similar cases with control clones had no visible effect (S1E Fig). Constitutive *Inx2* RNAi in the somatic lineage completely blocked oogenesis and precluded follicle observations. Therefore, we added a constitutive *Gal80^{ts}* transgene to allow a temporal control of RNAi construct (18 °C then 48 h at 30 °C). Inx2 knockdown induced by RNAi in follicle cells during 48 h leads to a strong reduction of the proportion of Edu-positive germline cysts, indicating that the growth defect is associated with an effect on endoreplication (Fig 1J). Because it is known that germline and somatic growth influence each other, we aimed to determine whether *Inx2* effect on germline growth was due to a cell-autonomous effect on somatic growth [18,22,24,26]. We analyzed follicle cell growth in smaller clones that do not induce germline defect to avoid feedback between tissues. In such mutant cells we did not observe any difference in cell size and in the proportion of cells in S phase (EDU-positive) between these populations, indicating that Inx2 did not influence somatic growth in a cell autonomous manner (Fig 1K, 1L). Of notice, follicle cells express other innexins at their lateral domain, suggesting that functional gap junction in between follicle cell may still exist in absence of Inx2 and may mask a potential communication in between these cells influencing their growth [28]. Nonetheless, these results suggest that Inx2-dependent gap junctions are specifically required for germline growth. We could not confirm this hypothesis using germline null *Inx4* mutant clones because in these mutant germline cysts development is blocked very early in the germanium [27]. Therefore, we generated *Inx4* RNAi clones in the germline, directly detected by the absence of Inx4 expression (Fig 1G). *Inx4* RNAi cysts were not larger than the younger wild-type follicle, indicating defective germline growth (Fig 1G, 1M). Altogether, these data indicate that Inx2 and Inx4 form homomeric and heterotypic gap junctions between somatic and germ cells that are required for germ cell growth.

## Genes implicated in amino acid metabolism are enriched in somatic follicular cells

Gap junction requirement for germ cell growth supports a model in which metabolites diffuse from follicle cells to germ cells. However, the direct identification of such metabolites is technically challenging. We hypothesized that if follicle cells produce or import metabolites not present in the germline, we might identify enzymes or transporters that are involved in this process and that are more expressed in follicle cells. To this aim, we performed translating

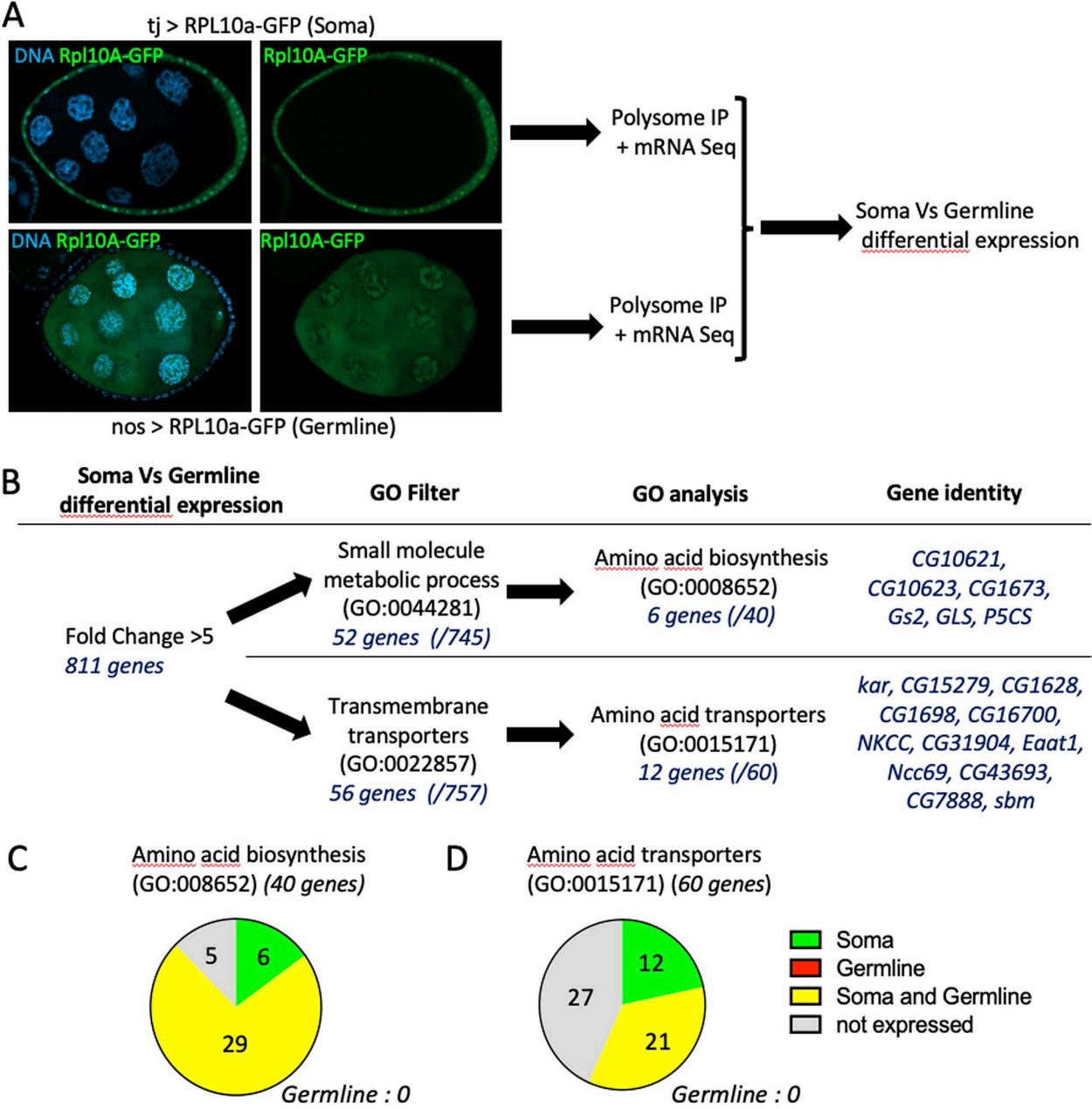

**Fig 2. TRAP analysis identifies soma-specific expression of amino acid metabolism and transport genes.** (**A**) Representative images of follicles expressing RPL10a-GFP in the soma (driven by tj:Gal4) or the germline (driven by nos:Gal4p16). After polysome immunoprecipitation (IP), mRNAs were sequenced. (**B**) The expression of 811 genes was significantly enriched in somatic cells compared with germline cells (fold-change > 5, *p*-value < 0.05). After filtering for genes involved in small-molecule metabolism or transport, genes related to amino acid biology were identified. (**C**) Distribution (soma and/or germline) of genes encoding proteins implicated in amino acid biosynthesis or transport according to their expression profile. No gene of these classes was enriched exclusively in the germline. The raw data underlying this figure can be found in S1 Table.

ribosome affinity purification (TRAP), an approach that allows identifying the tissue-specific translatome. We used *nanos:Gal4VP16* and *trafficjam:Gal4* drivers to express a Green Fluorescent Protein-tagged ribosomal protein (UASp:RPL10a-GFP) specifically in the germline and in somatic cells, respectively (Fig 2A). Then, we immunoprecipitated the GFP-tagged polysomes and isolated and sequenced the associated mRNAs under translation. Our biological

replicates were highly reproducible (S1 Table and S2A Fig). To extract genes with a specific or strongly enriched somatic expression, we used a fold-change of 5 between soma and germline as cut-off that gave a list of 811 genes (Figs 2B and S2B). We then selected genes involved in small molecule metabolic processes, reducing the list to 52 genes. The whole ecdysone synthesis pathway (six genes), which is exclusively active in follicle cells, was enriched in follicle cells, thus validating our TRAP approach [34]. Moreover, six genes encoding enzymes involved in amino acid biosynthesis were strongly enriched in follicle cells (Fig 2B). By analyzing this gene ontologyclass, we found that 29 enzymes were expressed in both cell types, but none was germline-specific (Fig 2C). Similarly, among the 60 amino acid transporters encoded by the fly genome, 21 were expressed in both tissues, 12 were specifically expressed in the soma, and none in the germline alone (Fig 2B, 2D). Interrogating the Fly Cell Atlas for the 18 genes involved in amino acid synthesis or transport and enriched in the soma in the TRAP experiments indicated that they were all detected in a higher proportion of "ovariole somatic cells" than "female germ cells" [35]. It suggests that the differences observed in the translatomes of soma and germline are already true at the level of their transcriptomes. These data indicated that among the genes strongly enriched in follicle cells there is a strong bias for genes involved in amino acid synthesis or import. Given the importance of these metabolites for growth, they are good candidates to be transferred through gap junctions.

## A putative amino acid transporter is required in follicle cells for germline growth

We performed a reverse genetic screen by inducing the silencing (RNAi) of the genes involved in amino acid synthesis or transport identified by TRAP in follicle cells and then looking for ovary growth defects. For each amino acid, many redundancies may occur between anabolic pathways or transporters that may preclude the observation of clear phenotypes. Moreover, although all the available RNAi lines for these genes were tested, we could not exclude that some were not efficient enough. However, the knockdown of one of the 18 tested genes, *CG43693*, reproducibly induced an ovary growth defect in independent RNAi lines (Fig 3A–3C). We named this gene cochonnet (coch) for reasons explained below. The ovary growth defect observed upon *coch* knockdown is associated with a change in follicle stage distribution, with the accumulation of the younger stages (1–9) and a depletion of the more mature ones (10–14) (Fig 3D). Although uncharacterized in *Drosophila*, Coch unambiguously belongs to the well-characterized SLC36 family of amino acid transporters, and was especially close to mammalian SLC36A1 and SLC36A4 that are involved in non-polar amino acids transport as proline, alanine, or tryptophan [36–38].

RNA-FISH indicated that *coch* was specifically expressed in follicle cells, as no expression was detected in germline cells, and checked for its effective knockdown by RNAi in the somatic lineage (Figs 3E, 3F and S3A). Moreover, we detected *coch* mRNA in all follicle cells and at all stages of oogenesis, and its expression progressively increased with the stage (Fig 3E). This mRNA was enriched at the apical side of the follicle cell, fitting with a previous observation that many mRNAs, including this one, are localized at this domain in follicle cells [39]. A role in importing amino acids from the hemolymph would require a cell membrane localization, while some amino acid transporters are specific to some intracellular vesicular compartments. Therefore, to determine Coch protein localization, we generated a GFP knockin with the *Minos*-mediated integration cassette (MiMIC) system to insert in-frame an EGFP-encoding exon [40]. This insertion was in the non-conserved N-terminal part common to five of the seven described *coch* isoforms. Importantly, it tags isoforms RA and RB that were expressed in follicle cells according to our TRAP mRNA sequence data. The line *coch-GFP* was homozygous viable and fertile without visible phenotype. Moreover, ovary

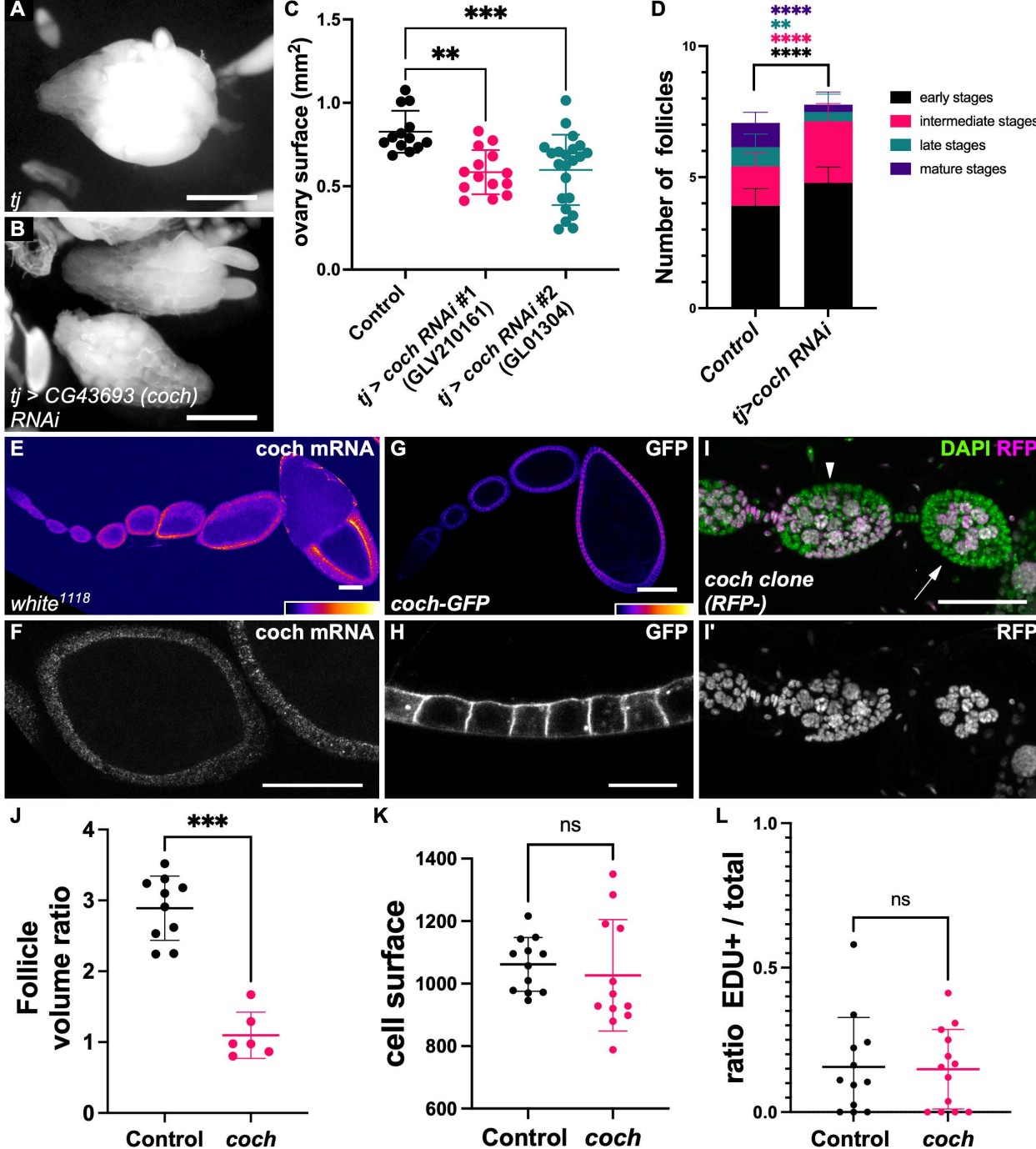

**Fig 3. Import of some amino acid in follicle cells promotes germline growth.** (**A**, **B**) Representative images of ovaries in (**A**) control and (**B**) after *CG43693/coch* knockdown by RNAi in somatic cells. (**C**) Ovary size (surface) quantification in control and after *coch* somatic knockdown using two RNAi lines (Ordinary one-way ANOVA + Dunnett's multiple comparisons test). (**D**) stage distribution per ovariole in control or *coch* knockdown conditions. We regrouped follicle stages in four categories: early: 1−6, intermediate: 7−9, late: 10−12, mature: 13−14 (*n* = 43 for controls and *n* = 95 for *coch* RNAi, two-way ANOVA and Šídák's multiple comparisons test). (**E**, **F**) In situ hybridization analysis of *coch* expression in (**E**) a whole ovariole and in (**F**) stage 4 and 7 follicles. (**G**) Expression of endogenous Coch protein tagged with GFP from germarium to stage 8. (**H**) Higher magnification of follicle cells showing Coch-GFP localization in the cell cortex. (**I**) *Coch* mutant clones marked by the absence of RFP (shown on I'). When all somatic cells of a follicle are mutated, follicle growth is affected (arrow) as seen when compared with the younger follicle (arrowhead). (**J**) Quantification of the ratio between the volume of follicle with a growth defect and fully covered by mutant cells and the *n* − 1 follicle in the same ovariole (*n* = 10 control and 6 *coch* mutant follicles, Mann–Whitney test). (**K**, **L**) quantification of (**K**) cell surface and (**L**)

EDU-positive cells in small *coch* mutant clones compared with the surrounding wild-type cells (*n* = 12 control and 12 mutant clones or groups of wild-type cells, Welch's *t*-test). For all graphs, data are the mean ± SD. **$p < 0.01$, ***$p < 0.001$. Scale bars: 500 μm in A, B, 50 μm D, E, F, H and 10 μm in G. The raw data underlying this figure can be found in S1 Data.

size was normal when *coch-GFP* was in trans with a deficiency covering the gene, indicating that the insertion did not affect protein function (S3B Fig). We observed a strong GFP signal that tended to increase throughout oogenesis in follicle cells, as previously observed for the mRNA, but no signal in the germline (Fig 3G). The GFP decorates the whole cortex of follicle cells, though appearing more enriched at the lateral and apical domains (Fig 3H). Of notice, septate junctions are not mature from stage 1–8, and the epithelium is therefore not impermeable, suggesting that solutes import via Coch can occur from any domain [41]. Moreover, Coch-GFP subcellular localization was not affected in mutant cells for *Inx2,* though its expression appears slightly lower in such a context (S3D Fig). Thus, both Coch tissue distribution, specific to somatic cells and present at all stages, and subcellular localization, at the cell cortex, were in agreement with the hypothesis that it is implicated in the import of some amino acids to promote follicle growth.

We then tested whether Coch was required for follicle growth. The Minos element insertion (*CG43693^{MI101960}*) used for the GFP knockin initially contains an exon with STOP codons in different frames. As it is located in the protein N-terminus, it should induce null mutation of the affected isoforms, which include the ones expressed in follicle cells. This allele was sublethal when homozygous or in trans with a deficiency covering the gene. In both cases, the ovaries of these flies were strongly atrophied, fitting with a role for this gene in follicle growth, though we could not detect a difference in the proportion of Edu-positive cysts (Figs S3B, 3C). Moreover, Replacing the STOP codon-containing exon with the GFP exon suppressed ovary growth defect. We generated mutant clones in the follicular epithelium. Analysis of small clones that did not cover the whole epithelium did not show any difference in cell size and proliferation, suggesting that follicle cell growth was not affected (Fig 3K, 3L). Conversely, when the whole epithelium of a follicle was mutated, such follicles were smaller (or of the same size) than the younger ones (Fig 3I, 3J). As this growth defect led to tiny and round follicles, we called this gene *cochonnet (coch)* after the nickname of the small wood ball used with larger metal boules for pétanque, a traditional game in the South of France. Importantly, these experiments indicated that *coch* expression in somatic follicle cells was required for germline growth.

## Genetic evidence of a metabolic transfer between soma and germline cells

Our results support a model in which some amino acids imported in follicle cells diffuses through gap junctions to sustain germline growth. In this case, germline expression of *coch* should rescue genetic conditions in which it is absent in somatic follicle cells. Importantly, it is known that the follicular epithelium is permeable at least until stage 8, and thus that metabolites from the hemolymph can directly reach oocyte surface [41]. To test this hypothesis, we combined the UAS/Gal4 system with the QUAS/QF system [42]. We generated a MatTub:QF driver and a QUASp promoter, inspired by the UASp promoter, for proper expression in the germline [43]. A Q*UASp:GFP* transgene driven by MatTub:QF was expressed once the germline cyst is formed in the germarium and in all the subsequent stages, starting slightly earlier than what described for MatTub:Gal4Vp16, indicating that both driver and QUASp promoter were functional (S4A Fig). Expression of the *QUASp:coch* transgene in the germline with the same driver (*MatTub>coch*) had a slight negative effect on ovary size (Fig 4A–4C, 4F). However, when it was combined with somatic knockdown of coch (tj>cochRNAi), it rescued

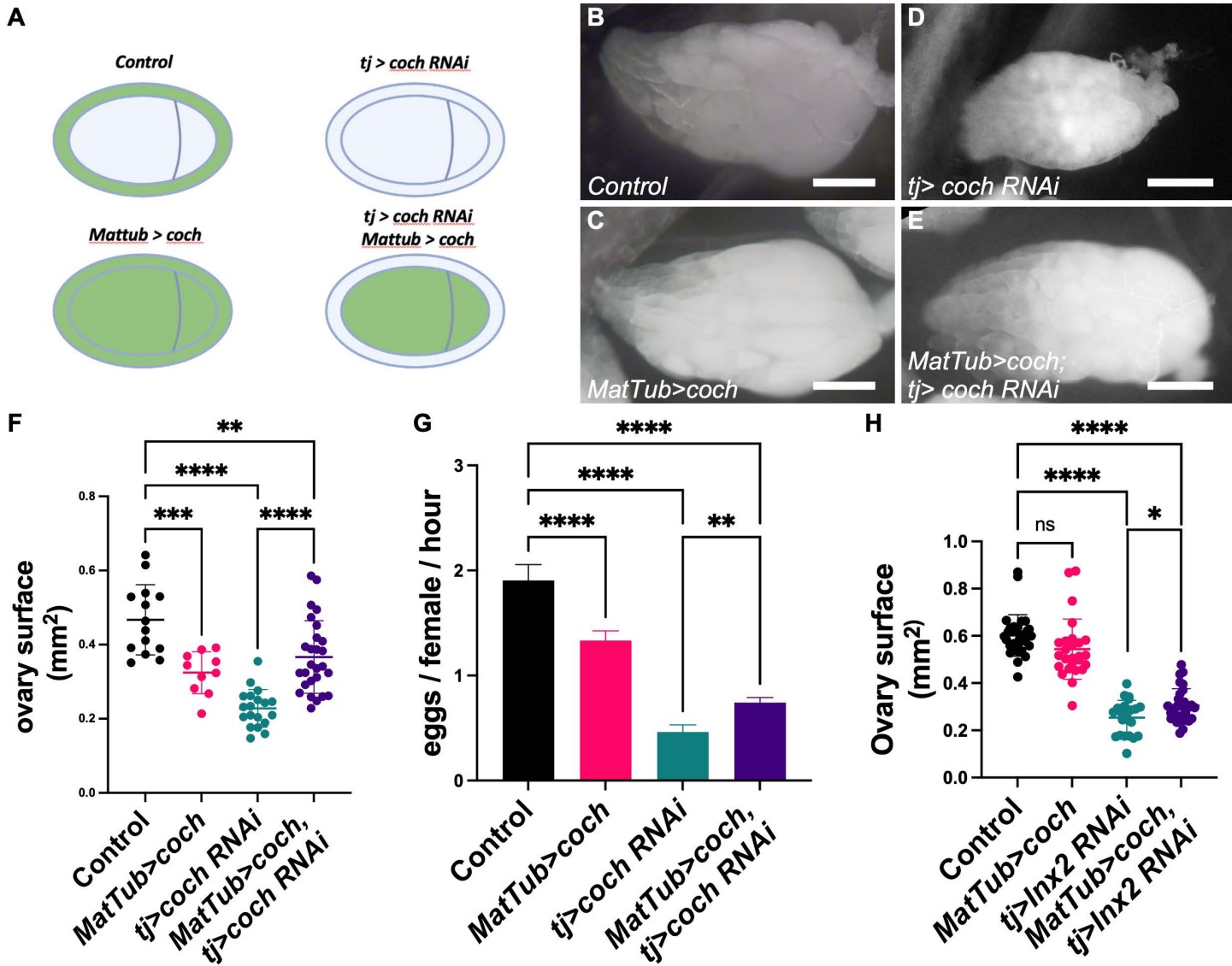

**Fig 4. Ectopic coch expression in the germline compensates its absence in the soma or the one of gap junction.** (**A**) Scheme representing *coch* expression in green in the different genotypes used on this figure. (**B–E**) Representative images of a full ovary from (**B**) a control female, (**C**) a *MatTub>coch* female, (**D**) a *tj>coch RNAi* female, a *MatTub>coch* female and (**E**) a *tj>coch RNAi*, *MatTub>coch* female. (**F**) Quantification of ovary size (surface) in the indicated genotypes ($n$ = 14, 10, 19, and 27 ovaries, Ordinary one-way ANOVA + Tukey's multiple comparisons test). (**G**) Quantification of egg laying per female and per hour, from four independent experiments with 10 females, for the indicated genotypes. (**H**) Quantification of ovary size (surface) in the indicated genotypes. As we pointed out, a slight variability between two independent experiments, but still with the same tendencies, data were analyzed using two-way ANOVA plus Šídák's multiple comparisons test ($n$ = 30, 26, 22, 30 ovaries). For all graphs, data are the mean ± SD., *$p$ = 0.0378, **$p < 0.01$, ***$p < 0.001$, ****$p < 0.0001$. Scale bars: 500 μm. The raw data underlying this figure can be found in S1 Data.

the effect of the latter on ovary size (Fig 4A, 4D–4F). Importantly, since MatTub:QF driver is expressed only from late stages of germarium, the observed rescue cannot be due to an effect during gonad development. However, such rescue is not quantitatively observed in the stage distribution, suggesting that the latter is less sensitive than ovary size to probe growth modification (S4B Fig). We also quantified egg laying in these different genetic backgrounds. This experiment confirmed both the slight deleterious effect of germline *coch* expression, but also its ability to rescue its knockdown in the follicle cells (Fig 4G). Altogether, these genetic

data demonstrated that the requirement of coch expression in follicular cells for germline growth can be partially compensated by its germline expression, suggesting a direct metabolic exchange between these cell types through the gap junctions. To further explore this idea, we attempted similar experiments between the ectopic expression of *coch* in the germline and the *Inx2* knockdown. Of notice, in the conditions used for these experiments (18 °C then 48 h at 30 °C), due to the temporal control of *Inx2* RNAi expression, we did not observe a significant negative effect of germline ectopic *coch* expression that we observed before (Fig 4F, 4H). However, it tends to rescue the ovary size of *Inx2* knockdown (Fig 4H). Interestingly, this result fits with a model in which the solutes imported via Coch in the somatic cells reach the germline through Inx2-dependent gap junctions.

## Blocking gap junctions or amino acid import induces processing bodies

Previous studies indicated that somatic growth impairment affects germ cell development [18,22–24]. Moreover, the mRNA binding protein Me31B, which is typically concentrated where the anterior–posterior axis determinants localize in the oocyte, becomes enriched in large cytoplasmic structures in the germline when somatic growth is impaired [16,44]. These condensates are reminiscent of P-bodies and stress granules observed upon various stresses, including amino acid deprivation [45]. In follicles, their formation can be induced by reducing protein availability in fly food and are more easily seen at stage 9 in Me31B-GFP-expressing follicles (Fig 5A, 5B) [44]. Therefore, we set up a semi-automated method for their quantification by measuring the fluorescence fraction found in condensates in stage 9 nurse cells (Fig 5C). P-body formation is usually induced by inhibitory phosphorylation of the eukaryotic translation initiation factor 2 subunit alpha (eIF2α) on serine 51 (S51), an event also known to cause a translation arrest [45]. Overexpression in the germline of a transgene mimicking this phosphorylated form (eIF2α-S51D), but not wild-type eIF2α, strongly induced P-body formation, although we did not quantify them because these follicles never reached stage 9 (Fig 5D, 5E). Importantly, eIF2α-S51D overexpression was sufficient to completely block germline growth, suggesting a link between P-body formation and growth control (Fig 5E). As P-bodies potentially repress germline growth, we tested whether the absence of gap junctions between germ cells and follicle cells or defective amino acid import in the follicle cells could induce their formation. We knocked down *Inx2* and *coch* in follicle cells using *tj:Gal4* in the presence of Me31B-GFP. To be able to recover and observe stages 8–9, *Inx2* RNAi was induced only during a short period of time (14 h, at 30 °C). We also attempted Inx4 knockdown by inducing RNAi specifically in the germline. *Coch*, *Inx2* and *Inx4* knockdowns strongly induced the formation of P-bodies when compared to their respective control, suggesting that the observed growth limitation was linked to their formation and a potential arrest of translation (Fig 5F–5L). We also noticed that P-body aspect may differ from a follicle to another, being more dotty or more flaky, though we could not find an explanation for this difference as it appeared not linked to the genotype or the amount of P-bodies. Finally, since ectopic *coch* expression in the germline tended to rescue ovary size in somatic *Inx2* knockdown, we asked whether a similar effect could be observed on P-bodies. Interestingly, we found a significant reduction in P-bodies compared to *Inx2* knockdown (Fig 5J). These results provide another indirect argument in favor of a model based on the exchange of solutes across gap junction that are initially imported by Coch in the somatic cells.

We hypothesized that the non-cell autonomous effect of *coch* and *Inx2* RNAi on the formation of P-bodies could a be consequence of the deprivation for some amino acids. P-bodies or stress granules formation in response to a lack of some amino acids can occur independently or dependently of eIF2α S51 phosphorylation. In the first case, which is not the most

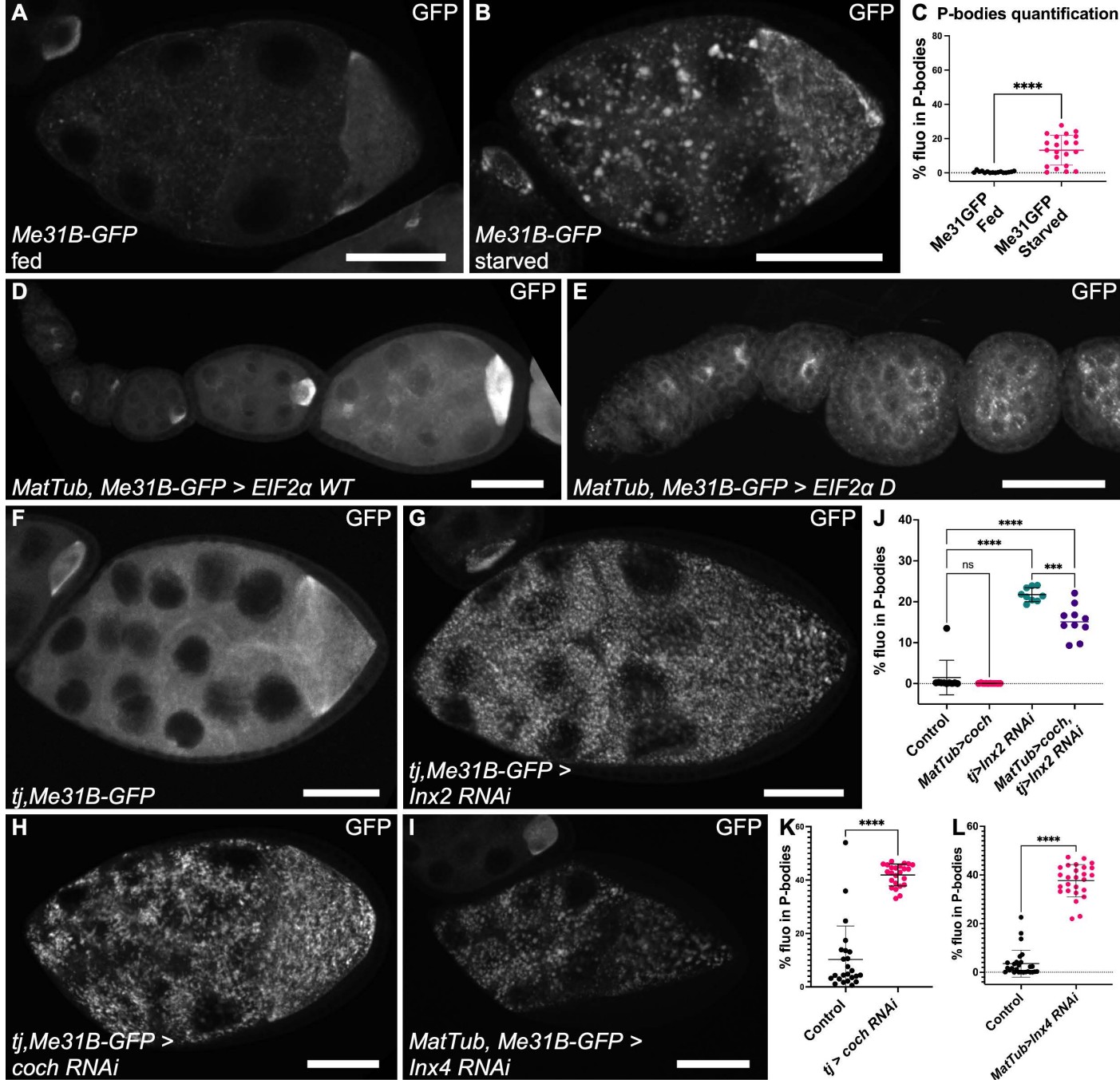

**Fig 5. Defective gap junctions or amino acid import induces P-body formation.** (**A**, **B**) Endogenous GFP-tagged Me31B protein expression in stage 9 follicles in (**A**) well-fed conditions or (**B**) after 15 h of protein starvation. (**C**) P-body quantification in normal and starved conditions ($n = 16, 21$). (**D**, **E**) Me31B-GFP expression in ovarioles germline that overexpresses (**D**) wild-type eIF2α or (**E**) or eIF2α-S51D (inhibitory phosphorylation). (**F–I**) Representative images of Me31B-GFP expression in stage 9 follicles in control (**F**) or after RNAi-based knockdown in somatic cells of (**G**) *Inx2*, (**H**) *coch* and (**I**) in the germ cells of *Inx4*. (**J**) P-body quantification in follicles from control females, *MatTub>coch* females, *tj>Inx2 RNAi* females, and *tj>Inx2 RNAi, MatTub>coch* females ($n = 10, 10, 9, 10$ follicles and one-way ANOVA plus Tukey's multiple comparisons test). (**K**, **L**) P-body quantification in follicles after RNAi-based knockdown in somatic cells of *coch* (**K**) and in germline cells of *Inx4* (**L**) (K, control $n = 24$, coch RNAi $n = 24$, L, control $n = 27$, I*nx4 RNAi* $n = 26$ and Mann–Whitney test). For all graphs, data are the mean ± SD, ***$p < 0.001$, ****$p < 0.0001$, Unpaired *t*-test. Scale bars: 50 μm. The raw data underlying this figure can be found in S1 Data.

frequently described, it is associated with the repression of the TOR pathway [46]. However, induction of *Tor* RNAi in the fly germline does not induce P-body formation, excluding this possibility [16]. In the second case, eIF2α is phosphorylated by GCN2 that acts as an indirect sensor of amino acid availability. We generated null mutant alleles for *gcn2* that were homozygous viable, as similar alleles described during the course of this project [47–49]. Protein deprivation in *gcn2* transheterozygous females still led to P-body formation in the germ cells (S5A–S5E Fig). Moreover, an indirect reporter of GCN2 activity and eIF2α–S51 phosphorylation (ATF4-GFP) showed no signal in the germline of wild-type flies, even after protein deprivation (S5F, S5G Fig). Lastly, we tested more directly *gcn2* role in the germline when amino acid import in follicle cells is impaired (*tj>cochRNAi*). However, *gcn2* mutant germline clones did not increase the growth rate of wild-type and *coch* RNAi follicles (S5H, S5I Fig). These results strongly argued against the implication of GCN2 as a germ cell growth repressor when the availability of the amino acids provided by Coch is reduced. Thus, being independent of TOR and GCN2, the mechanism leading to P-bodies formation in the female germline seems unusual and will require further investigation.

Altogether these data indicate that although the two well-established pathways involved in amino acid sensing do not seem implicated, P-body formation acts as a metabolic stress sensor linked to the control of germ cell growth by gap junctions and the import of some amino acids in follicle cells.

## Gap junction assembly links intrinsic growth control and systemic control

Published data indicated that InR/PI3K or Tor pathway inhibition in follicle cells induces the formation of P-bodies in the germline [16]. This result was reproduced by *akt* silencing (RNAi), an essential actor of this pathway, in follicle cells (Fig 6A–6C). Since somatic activity of the InR/PI3K pathway also strongly influences germline growth, we asked whether there was a link between the InR/PI3K pathway activity in follicle cells and their ability to transfer metabolites to the germline. Coch-GFP expression level and localization were similar in wild-type and in follicle cells with a PI3K gain of function or mutated for *akt,* suggesting no impact on this specific actor of amino acid import in follicle cells (S6A, S6B Fig). Conversely, in *akt* mutant cells, Inx2 was almost undetectable (Fig 6D), whereas Inx2 expression was strongly increased in *Pten* mutant cells and plaques size was increased (Fig 6E). These data indicated that Inx2 expression is sensitive to InR/PI3K pathway gain and loss of function. Moreover, we observed that Inx2 protein level was also sensitive to the loss but not the gain of function of the Tor pathway, suggesting that it is not equally controlled by all the pathways modulating cell growth (S6C, S6D Fig). We also observed that *Inx2* mRNA level was strongly increased in *Pten* mutant follicle cells, suggesting that the observed effects on protein level and plaque assembly were due *Inx2* gene expression upregulation (Fig 6F). These observations supported a model in which the non-cell autonomous effect of the InR/PI3K pathway from somatic cells to germ cells is mediated by gap junctions. To test this hypothesis, we performed an epistasis experiment. As previously described [18], large *Pten* mutant clones in the follicular epithelium accelerated germline growth in a non-cell autonomous manner (Fig 6G). This effect was abrogated upon *Inx2* RNAi induction in *Pten* mutant cells (Fig 6H). Thus, gap junctions are required for germline growth control via InR/PI3K activity in the follicular epithelium. We also tested whether *Inx2* overexpression could be sufficient to rescue the knockdown of the InR/PI3K pathway in the somatic cells. In this genetic combination, we observed neither an increase in ovary size nor a decrease in P-bodies compared to the *akt* knockdown (S6E, S6F Fig). Thus, the control of Inx2 expression is not sufficient to explain the non-cell-autonomous effect on germline growth of the InR/PI3K pathway in the

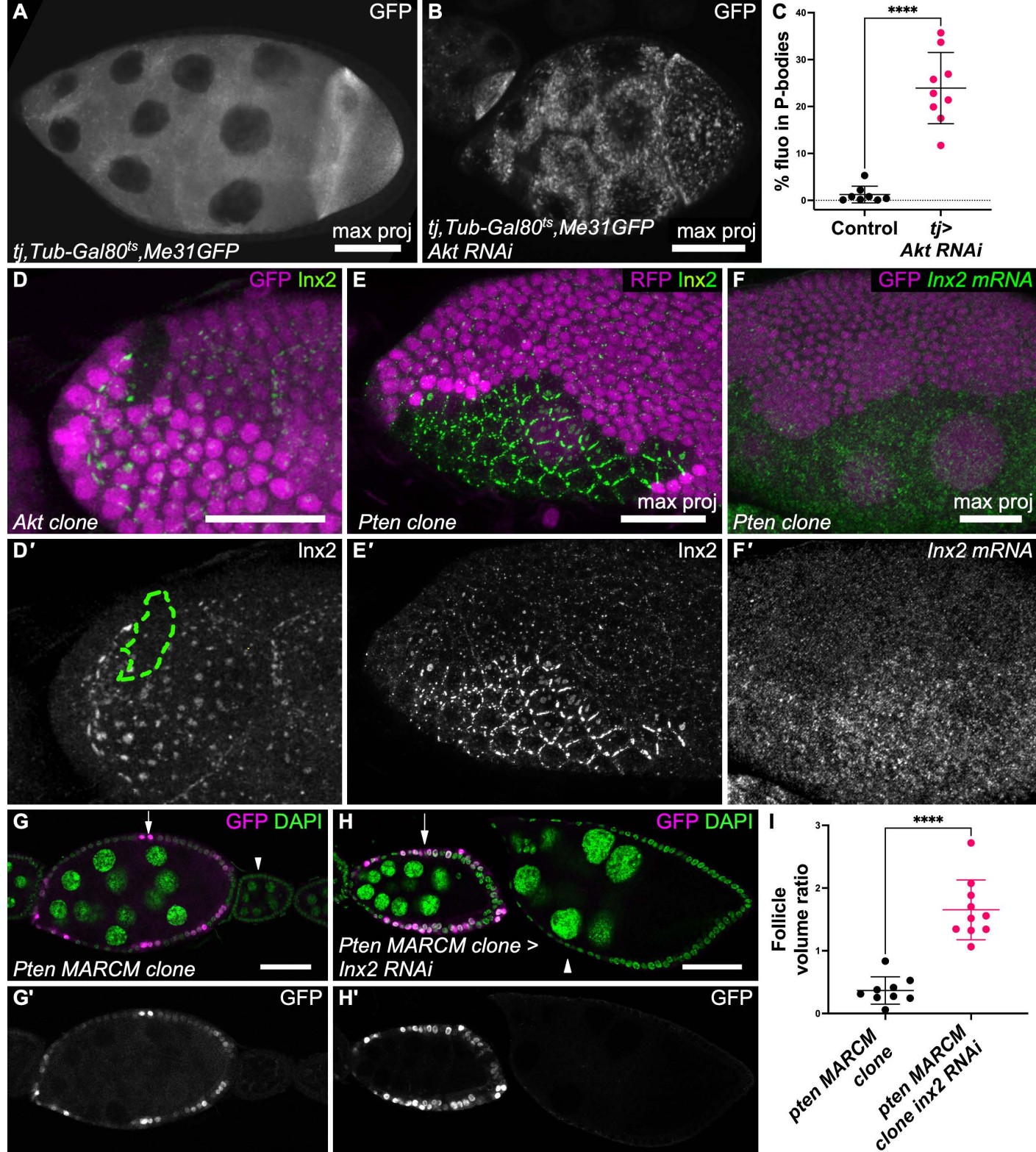

**Fig 6. Gap junctions are controlled by the InR/PI3K pathway.** (**A**, **B**) Representative images of Me31B-BFP expression in stage 9 follicles from (**A**) a control female and (**B**) a *tj:Gal4, Tub:Gal80*ts>*akt RNAi* female. (**C**) P-bodies quantification (fluorescence intensity) in the indicated genotypes (*n* = 8 and 9 follicles, unpaired *t*-test). Data are the mean ± SD. ****$p < 0.0001$. (**D**) Inx2 expression in an *akt*q mutant clone (marked by the absence of GFP expression). (**E**, **F**) Inx2

protein (**E**) and *Inx2* mRNA expression by FISH (**F**) in a *Pten*dj189 mutant clone marked by the absence of RFP expression. (**G**) Large MARCM *Pten*dj189 mutant clone marked by the presence of GFP expression showing faster follicle growth. (**H**) Large MARCM *Pten*dj189 mutant clone in which faster follicle growth was abolished after expression of an RNAi against *Inx2*. (**I**) Quantification of volume ratio between the older fully wild-type follicle and follicles containing a majority of *Pten* mutant cells and expressing or not RNAi against *Inx2* and (n = 10 and 9, Mann–Whitney test). Scale bars: 50 μm in A, B, G, H and 20 μm in D, E, F. Data are the mean ± SD. ****$p < 0.0001$. The raw data underlying this figure can be found in S1 Data.

follicle cells. Nonetheless, altogether, our data link the systemic control of somatic cells to the growth coordination between somatic and germline cells via the modulation of gap junction assembly.

## Discussion

In this article, we show that gap junctions participate in the control of cell growth, leading to the intuitive proposal that they allow a metabolic flow between cells. In accordance with this hypothesis, we identified a putative amino acid transporter, *coch*, specifically expressed in follicle cells and required for germline cell growth. Though a flux of amino acids from somatic cells to germ cells cannot be directly visualized, *coch* expression in the germline can partially bypass its silencing or the one of *Inx2* in the soma, suggesting a functional metabolic exchange between these cell types through gap junctions. Moreover, direct genetic induction of P-bodies in the germline is sufficient to block germline growth, and both defective gap junction or amino-acid import in somatic cells induces P-bodies in the germ cells, a feature usually associated with an arrest of translation. However, this effect is not mediated by GCN2 and it is therefore unclear by which mechanism P-bodies are induced in such a situation. Thus, the most plausible explanation for all our data is a model in which gap junctions allow the transfer from somatic cells to germ cells of metabolites, such as amino acids, that germ cells cannot directly produce or import, thereby promoting germ cell growth through translation control.

Inx2 and Inx4 have various functions during *Drosophila* oogenesis [6,27,29,31]. However, our data showing their involvement in germ cell growth can be more easily compared with gap junction role in mammal follicles where Cx37 is expressed in the oocyte and Cx43 in follicle cells [7,8]. This indicates that gap junction requirement for oocyte growth is a conserved feature throughout evolution. Gap junctions are present in most animal tissues and can directly connect different cell types, for instance, neurons and glial cells. Therefore, their involvement in growth control might be more general. We found that Inx2 expression and the subsequent formation of plaques were strongly regulated by the InR/PI3K pathway in a cell autonomous manner, providing an effective mechanism to link metabolic flow with growth systemic signals. Since InR/PI3K pathway activity in the follicle cells controls both somatic growth and germline growth ensuring their coordination, its impact on gap junctions likely participates in this coordination.

Our results also established that Coch, a putative amino acid transporter of the SLC36A family, must be expressed in somatic cells for germline growth. Amino acid availability and cell growth control are usually linked by the TOR pathway [50]. However, this pathway is unlikely implicated in the mechanism described here. Indeed, first, when germline growth is blocked due to alteration of somatic cell growth upon loss of *akt*, which also regulates gap junction assembly, the TOR pathway is still active in the germline [18]. Moreover, overactivation of the TOR pathway in the germline in such conditions does not suppress growth inhibition. Finally, TOR inhibition in the *Drosophila* female germline does not induce P-body formation [16]. Therefore, we tested the involvement of the other well-described amino acid sensing pathway that relies on GCN2. Our results also excluded this mechanism to explain P-body formation and growth inhibition in the absence of proline import in somatic cells or

of gap junctions. Therefore, more studies are needed to determine the precise mechanism underlying germline growth control.

Several arguments suggest that amino acids imported by Coch might just be the tip of the iceberg of the metabolic cooperativity between follicle cells and germ cells. *Inx2* mutant clones have a stronger impact on germline growth than *coch* mutant clones. Probably as a consequence of this difference of requirement, temporally controlled Inx2 knockdown induces an acute and strong arrest of germline growth, associated with a cell cycle arrest, while *coch* mutation has a milder effect that may lag each cell-cycle phase, but then do not significantly affect their respective proportion. In agreement with this idea, when somatic clones for *coch* induce smaller follicles, it is also associated with smaller nurse cell nuclei, which strongly suggests a delay in the cell progression through the rounds of endoreplication. Thus, it suggests that other metabolites from follicle cells also may promote germline growth. Accordingly, our translatome analysis showed that different genes involved in amino acid synthesis or import were specifically expressed in follicle cells, but none in germ cells. The absence of phenotype when these genes were knocked down, except *coch*, could be explained by multiple potential levels of redundancy among synthesis pathways, transporters, and synthesis and import. Nonetheless, detailed analysis of the genes overexpressed in follicle cells compared with germ cells suggest that several amino acids (e.g., proline, tryptophane, valine, tyrosine) could flow from the soma to the germline. Moreover, in the mouse, the alanine transporter SLC38A3 is strongly enriched in granulosa cells that are required for efficient alanine import in the oocyte [11]. Thus, altogether these data suggest that gap junction-dependent amino acid flow is not restricted to the amino acids imported via Coch. Besides, glucose intake and pyruvate production are more efficient in granulosa cells, suggesting that follicle cells could provide energetic molecules to the oocyte [12]. In line with this observation in mammals, TRAP data indicated that transporter for trehalose (tret1-1), the circulating sugar in insects, was expressed in somatic cells and not in the germline, whereas germline development is highly dependent on sugar and the pentose phosphate pathway [51]. Thus, a contribution of energetic metabolism in the soma–germline cooperativity via gap junctions will be an interesting avenue for future investigations. Moreover, it has already been shown that calcium can be exchanged between germ cells and follicle cells by Inx2/Inx4 gap junctions, and a potential role in the control of germ cell growth of this ion that can act a second messenger cannot be currently excluded [29]. Finally, the fact that Inx2 expression in follicle cells is not sufficient to explain the non-cell-autonomous effect on germline growth of the InR/PI3K pathway in the follicle cells, strongly suggests that this pathway controls other genes involved in the exchanges between the two tissues.

One might ask what is the evolutive advantage of this metabolic cooperativity compared with direct import or synthesis in the germline. Three main hypotheses could be proposed. First, metabolites might not be able to directly reach oocyte membrane due to the presence of the follicular epithelium. However, in follicles from stage 1–8, septate junctions are immature, and the epithelium is not impermeable [41]. Accordingly, our results showing a partial rescue of *coch* absence in the soma by its ectopic expression in the germline imply that its solutes can reach the oocyte. Second, female germ cells undergo massive growth and there is a non-linear relation between volume and cell surface increases. Consequently, surface exchange with the extracellular medium might not allow sufficient metabolic import and may require the support of follicle cells. This hypothesis was proposed to explain gap junction requirement for mammal oocyte growth [52]. It could also explain why germline growth, but not somatic growth, is sensitive to the loss of *coch*, as follicular epithelium has a relatively constant height and thus increases much less in volume than the germline. However, in this case, both tissues should be able to import the required metabolites and our data does not support such a model

because follicle cells expressed a whole set of anabolic enzymes and transporters that were not expressed in the germline. Alternatively, such a mechanism may provide a protective effect for the germline. It is well established that the cell metabolic activity can induce stress, with for instance, the production of reactive oxygen species (ROS). However, the germline must be protected, especially its DNA content that is transmitted to the embryo. A recent study demonstrated that mammalian oocytes block ROS production by suppressing mitochondrial complex I [53]. Thus, a gap junction-mediated metabolic exchange might allow externalizing the stress-generating metabolic activity to follicle cells that will anyway die few days later. Notably, *coch* ectopic expression in germ cells had a slight but significant negative impact on ovary size and egg laying, suggesting that its presence in germ cells is detrimental for follicle development. Although, the exact reason for this deleterious effect is unknown, this observation fits with a model in which externalization of metabolic import and activity could facilitate proper oocyte development. However, Coch is a transporter, and not an enzyme, and thus, the protective effect might not be directly due to amino acid exclusion, but to one of the many possible downstream metabolic activities.

Altogether, our data indicate that gap junctions and a metabolic flow are essential for cell growth and that gap junction assembly can be modulated to adjust cell growth rate. Moreover, it suggests that amino acids, and potentially other metabolites, are important actors in this mechanism ending with the formation of P-bodies, opening a large field for investigation to obtain a comprehensive view of this metabolic cooperativity.

## Materials and methods

### Fly genetics and handling

All fly stocks used are detailed in S2 Table. The final genotypes, temperature and heat-shock conditions are in S3 Table. *gcn2* null alleles were generated by inducing indel mutations using an available gRNA line. Alleles are described in S2 Table. Unless specified, flies were kept on a corn meal-based medium with 80 g/L fresh yeast. Protein starvation was performed on grape juice agar plate. Coch-EGFP was obtained by inserting the EGFP-FlAsH-StrepII-TEV-3xFlag cassette in the Minos element insertion MI01960, as previously described [40]. For egg laying quantification, 10 females of each genotype were place on fruit juice agar plate with liquid yeast for few hours, then the number per hour and per female was calculated and the experiments were repeated four times.

### TRAP experiments

Ovaries of 100 females of each genotype (tj>RPL10-GFP or Nos>RPL10-GFP) were dissected on ice. Then, TRAP was performed as described in [54]. Briefly, after homogenization, ovary extracts were preabsorbed with magnetic beads. Immunoprecipitation was performed with anti-GFP antibodies already coupled with magnetic beads before mRNA extraction. Tissue specificity and enrichment of the extracted mRNAs were checked by Reverse Transcription-qPCR with *GFP* (enrichment), *traffic-jam* (soma specific), *Ago3* and *Aubergine* (germline specific) primers. mRNA libraries were made with the Nugen Ovation 1–16 droso Universal RNA-seq kit according to the manufacturer's instructions. Sequencing was performed by Fasteris. Data were deposited on GEO (GSE230452) and described in S1 Table.

### Molecular cloning and transgenesis

The *QF* sequence was amplified from the pAttB-QF-sv40 vector and cloned in the vector that contains the alpha4-tubulin promoter (MatTub). The pQUASp vector was constructed

from pUAST in which the promoter was replaced by QUAS sites and the minimal P-element promoter was amplified from the pUASp vector. Then, this vector was used to clone *EGFP* or *coch* coding sequence and P-element insertions were generated. *eIF2α* and *eIF2α-S51D* coding sequences were cloned in the pUASz vector, and transgenes were inserted at the AttP40 landing site. All vectors and new *Drosophila* lines can be provided upon request.

## Immunostaining, FISH, EDU incorporation, imaging and quantitative analyses

Resources and reagents are listed in S2 Table. Immunostaining was performed as described in Vachias and colleagues (2014). Stellaris SM-FISH oligo-probes against *Inx2* and *CG43693/coch* mRNAs were produced by Biosearch Technologies. FISH was performed according to the manufacturer's instruction. Images were acquired on a Zeiss LSM800 confocal microscope or a Zeiss Cell observer spinning disc microscope. P-bodies were quantified using a homemade macro initially designed to quantify basement membrane fibrils [55]. Quantifications were done on 5 μm z-stack projections of spinning-disc images acquired with a ×20 lens. After manual selection of a ROI corresponding to nurse cells, P-bodies were detected by keeping objects with a minimal size of 5 adjacent pixels and a minimal fluorescence intensity of ×1.75 of the mean intensity. Then, the total fluorescence contained in these P-bodies was quantified and reported as a fraction of the total fluorescence of the ROI. For EDU incorporation, ovaries were incubated with 10 μM EDU in complemented Schneider medium for 15 min. After fixation, staining was performed according to the manufacturer's instructions (Kit EDU C10638, Thermo Fisher). Mutant-control follicle volume ratios were calculated after estimating the follicle volume as a spheroid ( $= 4/3\pi(\text{length} \times \text{width}^2)$ ). Mutant and control follicles in position 2–4 of the ovariole starting from the anterior were analyzed. Cell size was automatically determined after cell segmentation using Tissue Analyzer [56]. Statistical analyses were performed with Prism. For all experiments, the minimum sample size is indicated in the figure legends. For each experiment, multiple females were dissected. Randomization or blinding was not performed. The sample normality was calculated using the D'Agostino and Pearson normality test. Statistical tests and size samples are indicated in figure legends. Figures were prepared using ScientiFig [57].

## Supporting information

**S1 Fig.** (**A**) Maximum intensity projection images of an ovariole after immunostaining for Inx2 (green in A, white in A′) and Inx4 (magenta in A, white in A″). Note the progressive increase of Inx2 expression. (**B**) Control stage 9 follicle (no clone) showing that GAP junction plaques are absent at oocyte-follicle cells interface. (**C**) Follicle containing a germline mutant clone for *Inx2* (absence of GFP white in C and C′) and showing no growth defect and no impact on gap junction plaques while Inx2 staining (green in C, white in C″) is lost in the somatic clone observed on the same follicle. (**D**) Somatic *Inx4* RNAi clone showing no impact on plaque formation (Inx4 staining green in C, white in C″). E) Control FRT19A RFP minus clone covering the whole epithelium of a follicle and showing no growth defect when compared to surrounding follicles. Scale bars 50 μm in A, 10 μm in all the other panels.
(PDF)

**S2 Fig.** (**A**) Color-coded graph representing Spearman correlation coefficients between the different TRAP samples. (**B**) Z-score hierarchical clustering heat map visualization. Only significantly differentially expressed genes are reported. Highly expression correlation can be noticed across condition replicates. The raw data underlying this figure can be found in S1 Table.
(PDF)

**S3 Fig.** (**A**) RNAi clone against CG43693/coch stained by FISH against CG43693/coch showing its disappearance from these cells, confirming probe specificity and RNAi efficiency. (**B**) Quantification of ovary size (mm$^2$) for the indicated genotypes ($n$ = 13, 14, 12, 9, One-way ANOVA plus Dunnett's multiple comparisons test). Data are the mean ± SD. ****$p$ < 0.0001. (**C**) quantification of stage 1–8 follicles positive or negative for Edu in the germ cells (GL) ($n$ = 85 for control and $n$ = 143 for *coch* mutant flies, Fisher's exact test). (**D**) Coch-GFP in a *Inx2* mutant clone (RFP negative cells). Scale bars 10 μm. The raw data underlying this figure can be found in S1 Data.
(PDF)

**S4 Fig.** (**A**) Ovariole expressing a QUASP:GFP transgene under the control of the Mat-Tub:QF driver. (**B**) Stage distribution per ovariole in indicated genotypes. We regrouped follicle stages in four categories: early: 1–6, intermediate: 7–9, late: 10–12, mature: 13–14 ($n$ = 43 for controls and $n$ = 95 for *coch* RNAi, $n$ = 58 for qUAS:coch and $n$ = 58 for cochRNAi, qUAS:coch, two-way ANOVA and Šídák's multiple comparisons test). The raw data underlying this figure can be found in S1 Data.
(PDF)

**S5 Fig.** (**A**, **B**) Representative images of Me31B-BFP expression in stage 9 follicles from a: (**A**) control female, (**B**) starved control female, (**C**) gcn2 transheterozygous mutant female, and (**D**) protein starved gcn2 transheterozygous mutant female. (**E**) P-bodies quantification (fluorescence intensity) in follicles of the indicated genotypes and conditions. Data are the mean ± SD. (**F**, **G**) Absence of Atf4-GFP protein expression used as a read-out of GCN2 activity in (**F**) normal and (**G**) protein-starved conditions (green in F and G. white in F′ and G′). (**H**, **I**) Ovarioles with germline *gcn2* mutant clones marked by the absence of RFP expression (magenta) and stained for F-actin (green) in (**H**) wild-type background and (**I**) in flies harboring *coch* RNAi in follicle cells. The raw data underlying this figure can be found in S1 Data.
(PDF)

**S6 Fig.** (**A**) Coch-GFP expressing follicle with akt mutant clones (RFP-negative cells). (**B**) Coch-GFP expressing follicle with flip-out clones that express a constitutively active form of PI3K (RFP- positive cells). (**C**, **D**) Inx2 staining in follicles containing a mutant clones for (**C**) *Tor* or (**D**) *Tsc1*. (**E**, **F**) quantification of (**E**) ovary size and (**F**) P-bodies in the indicated genotypes (E: $n$ = 28, 19, 18, 32 and One-way ANOVA plus Tukey's multiple comparisons test, F: $n$ = 10, 10, 10, 9 Kruskal Wallis test plus Dunn's multiple comparisons test). For all graphs, data are the mean ± SD, ***$p$ < 0.001, ****$p$ < 0.0001. The raw data underlying this figure can be found in S1 Data.
(PDF)

**S1 Table. Excel file of soma versus germline TRAP analysis.**
(XLSX)

**S2 Table. Reagents and resources.**
(DOCX)

**S3 Table. Genotypes and specific conditions. (h: hours; HS: heat-shock, GP: grape juice agar plate).**
(DOCX)

**S1 Data. Numerical data supporting figures (excepted Figs 2 and S2).**
(XLSX)

## Acknowledgements

We are grateful to G. Junion, P Phelan, G Tanentzapf, H.D. Ryoo for sharing antibodies or fly stocks. We also thank the CLIC facility (Clermont Imagerie Confocale).

## Author contributions

**Conceptualization:** Caroline Vachias, Emilie Brasset, Vincent Mirouse.

**Data curation:** Yoan Renaud.

**Formal analysis:** Caroline Vachias, Vincent Mirouse.

**Funding acquisition:** Vincent Mirouse.

**Investigation:** Caroline Vachias, Camille Tourlonias, Louis Grelée, Parvathy Venugopal, Vincent Mirouse.

**Methodology:** Nathalie Gueguen, Parvathy Venugopal, Graziella Richard, Pierre Pouchin, Emilie Brasset.

**Project administration:** Vincent Mirouse.

**Software:** Pierre Pouchin.

**Supervision:** Caroline Vachias, Emilie Brasset, Vincent Mirouse.

**Visualization:** Yoan Renaud.

**Writing – original draft:** Caroline Vachias, Vincent Mirouse.

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
