## [Editor Report · Decision Letter 0]

8 Feb 2024

Dear Dr Mirouse,

Thank you for submitting your manuscript entitled "Gap junction and amino acid import in somatic cells promote germ cell growth" for consideration as a Research Article by PLOS Biology.

Your manuscript has now been evaluated by the PLOS Biology editorial staff as well as by an academic editor with relevant expertise and I am writing to let you know that we would like to send your submission out for external peer review.

Once your full submission is complete, your paper will undergo a series of checks in preparation for peer review. After your manuscript has passed the checks it will be sent out for review. To provide the metadata for your submission, please Login to Editorial Manager (https://www.editorialmanager.com/pbiology) within two working days, i.e. by Feb 12 2024 11:59PM.

Kind regards,

Ines

Ines Alvarez-Garcia, PhD

Senior Editor

PLOS Biology

---

## [Decision Letter · Decision Letter 1]

26 Mar 2024

Dear Dr Mirouse,

Thank you for your patience while your manuscript entitled "Gap junction and amino acid import in somatic cells promote germ cell growth" was peer-reviewed at PLOS Biology. Please also accept my apologies for the delay in providing you with our decision. Your manuscript has been evaluated by the PLOS Biology editors, an Academic Editor with relevant expertise, and by three independent reviewers.

The reviews are attached below. As you will see, the reviewers find the conclusions potentially novel and interesting, but they also raise several concerns and suggest several experiments that should be performed to convincingly support the main conclusions. Reviewer 1 thinks that some of the claims should be toned down and that it would be nice to know what aspects of growth are affected in the germline (ie. endoreplication), among other issues. Reviewer 2 mentions that the experiment showing metabolic coupling between soma and germline should be improved and suggests several experiments to confirm the findings. Reviewer 3 believes you should confirm the model proposing that Inx2-Inx4 mediated amino acid exchange between somatic cells and germline cells supports oocyte growth, explore the role of calcium in the process and check if the structural component or channel function of the gap junction is responsible for the observed growth defect, among other issues.

Based on the comments and following discussion with the Academic Editor, it is clear that a substantial amount of work would be required to meet the criteria for publication in PLOS Biology. However, given our and the reviewer interest in your study, we would be open to inviting a comprehensive revision of the study that thoroughly addresses all the reviewers' comments. The only exception is that direct evidence for small molecule (aa) transfer is hard to validate, but at least we would like to see somewhat indirect evidence – for example, Coch functioning in which cells, and leading to what phenotype, suggesting the 'most plausible explanation' as a model.

Given the extent of revision that would be needed, we cannot make a decision about publication until we have seen the revised manuscript and your response to the reviewers' comments. Your revised manuscript would need to be seen by the reviewers again, but please note that we would not engage them unless their main concerns have been addressed.

We appreciate that these requests represent a great deal of extra work, and we are willing to relax our standard revision time to allow you 6 months to revise your study. Please email us (plosbiology@plos.org) if you have any questions or concerns, or envision needing a (short) extension.

**IMPORTANT - SUBMITTING YOUR REVISION**

3. Resubmission Checklist

a) *PLOS Data Policy*

b) *Published Peer Review*

d) *Blurb*

Please also provide a blurb which (if accepted) will be included in our weekly and monthly Electronic Table of Contents, sent out to readers of PLOS Biology, and may be used to promote your article in social media. The blurb should be about 30-40 words long and is subject to editorial changes. It should, without exaggeration, entice people to read your manuscript. It should not be redundant with the title and should not contain acronyms or abbreviations. For examples, view our author guidelines: https://journals.plos.org/plosbiology/s/revising-your-manuscript#loc-blurb

Sincerely,

Ines

Ines Alvarez-Garcia, PhD

Senior Editor

PLOS Biology

Reviewers' comments

Rev. 1:

The manuscript by Vachias et al. examines a potential role for gap junctions in coordinating growth between somatic and germline cells during oogenesis by allowing transfer of metabolites. Previous studies in the mouse showed that gap junctions between somatic granulosa cells and the oocyte are required for oocyte growth and that metabolic coupling between these cell types is required for oocyte growth. Growth of the somatic and germline cells within the Drosophila ovarian follicle is also coordinated as shown by this group and others. Gap junctions have been visualized between the somatic follicle cells and between follicle cells and the ooctye/nurse cell complex and, although not acknowledged here, have been shown to permit communication between the follicle cells (Ref 28). However, whether they also coordinate somatic and germline growth in the Drosophila ovary and what metabolites are transferred isn't known. Although visualizing metabolite transport is not feasible in this system, the authors use alternative and clever approaches to provide evidence that the Drosophila ovarian follicle cells influence germline growth and that they do this at least in part by supplying amino acids whose transporters are not expressed in the germline. The data are convincing and for the most part, the conclusions are well supported. As discussed below, the assertions that gap junctions coordinate growth and that gap junctions ultimately regulate translation in the germline are less convincing. The work is strong enough to rest on the data provided, so the authors simply need to temper these assertions.

The authors show that heteromeric gap junctions form between the somatic and germline cells in the Drosophila ovary whereas homomeric junctions form between follicle cells. Using RNAi and genetic mutations, the authors show convincingly that gap junctions are indeed required for germline growth, but not for follicle cell growth. The finding that they are not required for follicle cell growth indicates that they nonautonomously influence germline growth, not that they coordinate follicle cell and germline growth (see the first concern above).

To identify metabolic pathways that use these gap junctions, they cleverly isolate mRNAs that are preferentially translated in follicle cells over the germline using TRAP. The data revealed differential production of proteins involved in amino acid synthesis or import. Although there are multiple ways to interpret this result, the authors focus the remainder of the study trying to support the idea that amino acids may be taken up by follicle cells and transported to the germline through gap junctions. RNAi knockdown of one of the genes identified, CG43693 (now named coch), a member of the SLC36 family of amino acid transporters, produced an ovary growth defect and both RNA and protein expression appear specific to the follicle cells. The authors nicely combine the QUAS and UAS systems and show that ectopic expression of coch in the germline can compensate for knockdown in the follicle cells. This supports the idea that the germline requires the activity of coch and amino acid transport and because coch normally only functions in the follicle cells. Although it would be ideal to have more direct evidence, the authors argue reasonably that the data support direct metabolic exchange between somatic and germline cells.

In the remainder of the manuscript, the authors try to connect metabolic transport to the translational state of the oocyte, which I find to be much less convincing. It is clear that P bodies do form when gap junctions are eliminated or when coch is knocked down, but whether this relates to depletion of amino acids is not clear. Me31B containing P bodies form/enlarge under many different stress conditions and it seems more likely that they are a consequence of metabolic stress (from any one of many possible metabolites not considered here) rather than a specific mechanism to stop translation. Therefore, the statement that although the statement "P-body formation is linked to germ cell growth control by gap junctions and amino acid import in follicle cells" the authors should make it clear that P-body formation may be a downstream consequence rather than direct effector of this growth control.

Specific issues:

1) Why are the follicle cells unaffected in coch mutants or when gap junctions are disrupted throughout the epithelium (so there is no exchange between follicle cells)?

2) The authors refer to amino acid transport generically but with coch knockdown/mutation, only a small subset of amino acids is not transported. The authors should be careful about implying that all amino acid transport is blocked. Moreover, transport from hemolymph to the oocyte can occur directly up to stage 8, and the authors show that some transporters are expressed in the oocyte. Finally, the 5-fold enrichment threshold used to designate gene expression as specific to the follicle cells doesn't mean that there is no expression in the oocyte and without RNAi/knockdown experiments as performed for coch it is premature to conclude that there is no expression of a particular transporter in the germline.

3) It would be nice to know what aspects of growth are affected in the germline. Other effectors of growth like myc affect nurse cell endoreplication (doi: 10.1242/dev.00932). Effects endoreplication are relatively easy to assay.

4) Knockdown of other genes identified in the TRAP experiment did not result in phenotypes and the authors suggest that there may be multiple levels of redundancy. Do they have evidence that the RNAi actually worked?

5) Figure 3D and E - the resolution is too low in these figures. At the resolution shown, the RNA looks like it is cortical, which makes the result questionable.

6) For the genes whose translation is limited to the follicle cells, it would be interesting to know if they are only transcribed there and not in the nurse cells, or if their translation is controlled so that they are only translated in the follicle cells. Are there RNA seq data available that could be used to answer this?

7) The hierarchical clustering shown in Fig. S1 is not a good way to show reproducibility of results. A graph with Pearson correlation coefficients is preferable.

8) Some figures are labeled with CG43693 and some with coch (even before the name is defined). Best to be consistent throughout or to use CG43693 in the early figures and coch after the name is defined.

Rev. 2: Mayu Inaba – note that this reviewer has signed her review

The article by Vachias et al demonstrates that gap junctions participate in metabolite transfer from somatic follicle cells to germline to support oogenesis. Using polysome-seq, which can reliably detect cell-type specific gene expression, authors identified an amino acid transporter, coch, specifically expressed in follicle cells but required for growth of germ cells. Coch knockdown phenotype in somatic FCs is rescued by ectopic expression of Coch in germline, supporting the proposed idea of metabolic exchange between these cell types. Overall, the authors perform well thought out experiments, which support a novel mechanism how somatic epithelia support oogenesis via sending amino acid through gap junctions. There are, however, some points to be consider worth revising for the overall improvement of this manuscript.

Major points:

1: As authors stated, metabolic coupling between soma and germline is solely supported by experiments shown in Fig4, in which they show coch expression in the germline can bypass its silencing in the soma. I feel this experiment needs to be done with better methods, such as checking frequency/distribution of stages of egg chambers.

2. Can inx4 RNAi in germline (or inx2 in soma) phenotypes be suppressed by expression of Coch in germline? I understand that there should be many other materials to be transferred through gap junctions, but I wonder if those amino acids could be essential for oogenesis, if so the rescue may occur even partially. On the other note, p-body formation seems to be a good lead-out and I wonder if authors can better quantify non-cell autonomy using this.

3. Can authors visualize inx2/4 patch in better resolution? In later vitelline membrane localize between germline and soma and I wonder if gap junctions are formed between microvilli.

4. The logic behind the experiments checking TOR and GCN2 function (Figure S5, line 256 to 270) is difficult to follow. It would be helpful for readers if authors can revise this portion. Especially, it is unclear why conclusion differ between previous studies and the current study.

5. Can inR effect or starvation on p body formation be cancelled by inx2 overexpression?

Minor points:

1. Quantification method of p-bodies are showing fluorescent intensity per detected p-body by taking minimum intensity of 1.75 compared with neighbouring pixels? If so, how authors quantified control?

2. fig 6D pattern of inx2 staining of surrounding wt FCs looks different from others.

Rev. 3:

The manuscript by Vachias et al have used the Drosophila oogenesis model to demonstrate that growth of the germline cells is dependent on the overlying somatic follicle cells. Employing classical genetics and by examining gap junction proteins, the authors claim that the communication through the gap junctions (Inx2 - Inx4) across soma and germline is critical for the growth of germ cells during oogenesis. In an effort to identify the nature of the molecule that may be passing through the Inx2-Inx4 heteromeric channel, the authors undertook a translatomic approach and narrowed down on a predicted amino acid transporter, CG43693. By depleting CG43693 function the authors claim that CG43693 is functioning in the follicle cells to support the germline growth. Finally by employing immunohistochemistry and epistasis experiments authors propose Inx2 functions down stream of Insulin signaling/ PI3K pathway in the follicle cells to facilitate germline growth. Overall the model proposed by the authors is quite novel but the primary claim needs further validation before it can be accepted. My comments are listed below.

Major comments:

1. The genetic evidence provided here doesn't unequivocally support the proposed model that Inx2- Inx4 mediated amino acid exchange between somatic cells and germline cells supports oocyte growth. This needs to be demonstrated more convincingly.

2. The function of CG43693 gene needs to be validated before it can be accepted as an as an amino acid transporter in the proposed model.

3. The aspect that Inx2- Inx4 colocalization is dependent on each other in developing egg chambers is a repeat of what has been shown by Sahu et al in 2021. How does adding similar data help in Fig 1.

4. Figure 1G, L- Did the authors check the status of sub apical Inx2 in the follicle cells when Inx4 was depleted from the germline cells?

5. Since the distribution of CG43693 is mostly apical and lateral in the follicle cells, it is difficult for one to understand as to how the aminoacids are internalized in the follicle cells? One would presume that uptake of amino acid may be facilitated from the basal side which is exposed to outside.

6. Since Inx2-Inx4 heteromeric channels have been previously demonstrated (Sahu et al 2021) to mediate calcium flow between follicle cells and nurse cells, have the authors checked if the calcium could play a role in this process?

7. As Inx2 is known to physically associate with DE-Cadherin and Armadillo (Bauer et al 2006), it will be important for the authors to check if the structural component or channel function of the gap junction is responsible for the observed growth defect. It will be worth checking the localization of CG43693- GFP in the follicle cells in Inx2 and Inx4 depleted backgrounds.

8. The authors are assessing the overall size of the ovary as readout for the germline growth. Since the size of the ovary can also be regulated by the number of ovarioles, it would be best to verify the results at the level of egg chambers (stage distribution and size) (Fig 3A-B, Fig 4).

9. Did the authors try to rescue the germ line growth defect induced Inx2RNAi by over expressing coch in the germline itself? Something along the lines of Fig 4.

10. The nature of the P bodies observed is different when Inx2 is down regulated in the follicle cell (Fig 5F) compared when coch RNAi is overexpressed in the follicle cell (Fig 5H). In Fig 5F the P bodies are punctate while in Fig 5H it seems more filamentous. Did the authors evaluate the P bodies when Inx4 was depleted in the germline cells.

11. Have the authors attempted to rescue the effects of coch-RNAi by supplementing the ovaries under live culture conditions with amino acids?

Minor comments:

1. How was the follicle volume calculated? The Materials & Methods section needs to be more elaborate.

2. How do the authors arrive at the statement that "Thus, both CG43693 tissue distribution and subcellular localization were in agreement with the hypothesis that it is implicated in the import of amino acids to promote follicle growth."

3. Could the TOR signaling in the follicle cells affect oocyte growth in the same ways as the authors observed for Inx2?

---

## [Decision Letter · Decision Letter 2]

11 Dec 2024

Dear Dr Mirouse,

Thank you for your patience while we considered your revised manuscript entitled "Gap junction and amino acid import in somatic cells promote germ cell growth" for publication as a Research Article at PLOS Biology. This revised version of your manuscript has been evaluated by the PLOS Biology editors, the Academic Editor and the three original reviewers.

Based on the reviews (attached below) and after discussing them with the Academic Editor and the rest of the team, we are likely to accept this manuscript for publication, provided you satisfactorily address the data and other policy-related requests stated below. While we do appreciate and acknowledge the comments made by Reviewer 3, after considering the other two reviews, we have decided to proceed without further revisions.

In addition, we would like you to consider a suggestion to improve the title:

"Gap junctions allow transfer of metabolites between germ cells and somatic cells to promote germ cell growth in the Drosophila ovary"

We expect to receive your revised manuscript by January 6. 

*Published Peer Review History*

*Press*

Sincerely,

Ines

--

Ines Alvarez-Garcia, PhD

Senior Editor

PLOS Biology

Fig. 1I-M; Fig. 3C, D, J-L; Fig. 4F-H; Fig. 5C, J-L; Fig. 6C, I; Fig. S2A, B; Fig. S3B, C; Fig. S4B; Fig. S5E and Fig. S6E, F

CODE POLICY

Reviewers' comments

Rev. 1:

The authors have adequately addressed my concerns.

Rev. 2:

Upon revision, the authors have remarkably addressed all issues pointed by reviewers, and I find it acceptable for publication.

Rev. 3:

In the revised manuscript by Vachias et al., the authors have tried to address the concerns of the reviewers to the maximum possible extent. In addition, they have tempered their claims to align with their findings. Though the experiments are nice, the interpretation of results might not be so straightforward. My concerns still remain, as we still lack mechanistic insight as to how Inx2 is regulating oocyte growth. Given that their findings are novel in the context of the role of Inx2 and CG43693 in oocyte growth, but due to technical challenges, the authors are not able to support their hypothesis conclusively. In light of this aspect, can this manuscript be considered under the 'Short Report' category?

Major comments:

1. Though the authors attempted to rule out the structural role of Inx2 in oocyte growth, the results were negative (Response to point no 7 for R3). In light of this result, it is difficult to accept that the channel activity of the Inx2 in the follicle cell is indeed modulating the germline growth.

2. The implication of amino acid downstream of Inx2 in oocyte growth is still speculative.

3. The function of the CG43693 gene is being proposed based on deep phylogenetic exploration. As this gene is an important part of this model, it would have been best to demonstrate its function under in vitro or in vivo conditions.

---

## [Editor Report · Decision Letter 3]

29 Jan 2025

Dear Dr Mirouse,

Thank you for the submission of your revised Research Article entitled "Gap junctions allow transfer of metabolites between germ cells and somatic cells to promote germ cell growth in the Drosophila ovary" for publication in PLOS Biology. On behalf of my colleagues and the Academic Editor, Yukiko Yamashita, I am delighted to let you know that we can in principle accept your manuscript for publication, provided you address any remaining formatting and reporting issues. These will be detailed in an email you should receive within 2-3 business days from our colleagues in the journal operations team; no action is required from you until then. Please note that we will not be able to formally accept your manuscript and schedule it for publication until you have completed any requested changes.

PRESS

Sincerely, 

Ines

--

Ines Alvarez-Garcia, PhD

Senior Editor

PLOS Biology
